# @CAUSALBENCH CHALLENGE 2023 - MINOR IMPROVEMENTS TO THE DIFFERENTIABLE CAUSAL DISCOVERY FROM INTERVENTIONAL DATA MODEL

## ABSTRACT

For the creation of new drugs, understanding how genes interact with one another is crucial. Researchers can find new potential drugs that could be utilised to treat diseases by looking at gene-gene interactions. Scale-based research on gene-gene interactions proved challenging in the past. It was necessary to measure the expression of thousands or even millions of genes in each individual cell. Recently, high-throughput sequencing technology has made it possible to detect gene expression at this level. These advances have led to the development of new methods for inferring causal gene-gene interactions. These methods use single-cell gene expression data to identify genes that are statistically associated with each other. However, it is difficult to ensure that these associations are causal, rather than simply correlated. So, the CausalBench Challenge seeks to improve our ability to understand the causal relationships between genes by advancing the state-of-the-art in inferring gene–gene networks from large-scale real-world perturbational single-cell datasets. This information can be used to develop new drugs and treatments for diseases. The main goal of this challenge is to improve one of two existing methods for inferring gene-gene networks from large-scale real-world perturbational single-cell datasets: GRNBoost or Causal Discovery from Interventional Data (DCDI). This paper will describe three small improvements to the DCDI baseline.

## 1 INTRODUCTION

Causal inference is a fundamental problem in science. Experiments are conducted in all fields of research to understand the underlying causal dynamics of systems. This is motivated by the desire to take actions that induce a controlled change in a system. However, studying causality in real-world environments is often difficult because it generally requires either the ability to intervene and observe outcomes under both interventional and control conditions, or the use of strong and untestable assumptions that cannot be verified from observational data alone.

To address these problem, CaualBench Chevalley et al. (2022) was introduced. CausalBench is a comprehensive benchmark suite for evaluating network inference methods on perturbational single-cell RNA sequencing data. It includes two curated, openly available datasets with over 200,000 interventional samples each, a set of meaningful benchmark metrics, and baseline implementations of relevant state-of-the-art methods. The CausalBench challenge also provides two different baseline methods for inferring causal relationship: the GRNBoost Aibar et al. (2017) and the DCDI Brouillard et al. (2020), and proposed changing one of the algorithms to improve its performance. The GRNBoost is a method for inferring gene regulatory networks from observational data. It can be improved by using interventional data, and the DCDI is a method for inferring gene regulatory networks from interventional data. It can be improved by tuning its parameters and by using more data. In this work, I chose to modify the DCDI baseline and apply three small modifications to the algorithm that are introduced in section 2.

## 2 METHODOLOGY

### 2.1 GREEDY PARTITIONING ALGORITHM

In the baseline implementation of the DCDI, the genes were partitioned into random independent sub-graphs, since DCDI can't handle the full graph as it does not scale well in terms of number of nodes. This partitioning scheme sacrifices possible causal links between genes in different sub-graphs to make the DCDI algorithm more tractable. So, to minimize the loss of any valid causal links, we need the partitioning algorithm to group the genes such that the genes in each sub-graph are related to each other as much as possible. The basic idea of the developed partitioning algorithm is to develop a measure of relationship between every pair of genes ($adj$ in the algorithm below), then after initializing the sub-graphs with random genes, we divide the genes into partitions using a greedy algorithm, i.e. a gene is assigned to a sub-graph where it has the maximum possible relationship with all other genes.

```
The Greedy partitioning algorithm:
            # initialize the algorithm parameters
            partition_length = int(len(indices) / self.gene_partition_sizes)
            indices = list(range(len(gene_names)))
            used = [False for i in range(len(indices))]
            random.shuffle(indices)
            # initialize the adjacency matrix
            adj = (expression_matrix >0).astype(int)
            adj = normalize(adj, norm='l2', axis=0)
            adj = np.matmul(np.transpose(adj), adj)
            # initialize partitions with random genes
            partitions = []
            for i in range(partition_length):
                partitions = partitions + [[indices[i]]]
                used[indices[i]] = True
            # divide the genes into partitions
            while not all(used):
                for i in range(partition_length):
                    if all(used):
                        break
                    max_dist, max_ind = -1, -1
                    for j in range(len(indices)):
                        if not used[indices[j]]:
                            dist = 0
                            for k in partitions[i]:
                                dist = dist + adj[k,j]
                            if dist > max_dist:
                                max_dist = dist
                                max_ind = indices[j]
                    partitions[i] = partitions[i] + [max_ind]
                    used[max_ind] = True
            # return the partitions
            return partitions
```

### 2.2 AUGMENTING THE DATA

In this work, the data is augmented to be the double of its original size. The augmentation algorithm is simple, you randomly select two samples with the same intervention, and average these two samples and add it as a new sample.

### 2.3 THE DEEP SIGMOIDAL FLOW MODEL PARAMETER TUNING

In the baseline model, the sigmoidal flow has two conditional layers with 15 dimensions each, and two flow layers with 10 dimensions each. However, it is a rule of thumb that each variable in a neural

network needs 25 examples to be trained well and to produce similar results across multiple runs, so the dimensions of the conditional and flow layers are set according to a simple heuristic with upper and lower bounds. The heuristic: $X = \sqrt{len(intervention)}/25/3/2/2/1.5$, and the dimension of the conditional layer is set to be $min(18, max(5, round(1.5 * X)))$, and the dimension of the flow layer is set to be $min(12, max(3, round(X)))$

## 3 CONCLUSION AND FUTURE WORK

In this work, three minor improvements to the DCDI baseline were introduced: new partitioning algorithm for the genes, data augmentation scheme and parameter selection formulas for the deep sigmoidal flow model. These modifications improved the performance of the DCDI baseline on the public test set. For future work, different measure of relationships between genes can be explored in the partitioning algorithm, also, a tractable more optimal partitioning algorithm can also be derived other than the proposed greedy algorithm.

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
