# OpenReview forum: "@CausalBench Challenge 2023 - Minor Improvements to the Differentiable Causal Discovery from Interventional Data Model"
_GSK.ai/2023/CBC_

### Official Review · Reviewer_ru7o · 2023-04-27
**Three improvements over a baseline method**

**Rating:** 6
**Confidence:** 4

**Review:**

Reviewer's summary:

This work proposes three improvements over the baseline DCDI algorithm: greedy partitioning algorithm, augmenting the data, and deep sigmoidal flow model parameter tuning.\
&nbsp;
&nbsp;

Reviewer's comments:

The report is concise and well-written. The proposed improvements are mainly based on heuristics and the rule of thumb. For example, fine-tuning the hyper-parameters of the flow model makes sense as it can be seen as a model selection criterion. However, the proposed method for data augmentation needs more elaboration. Data augmentation methods highly depend on the nature of data and its distribution. An effective augmentation would produce novel data points that come approximately from the same distribution as the original data and, at the same time, are not close replicates of existing data points. The report would significantly improve if the authors can explain why averaging is a valid augmentation for gene expression data and how effective it is as it's essentially a linear function of two existing data points.